# Semantic Uncertainty: Linguistic Invariances for Uncertainty Estimation in Natural Language Generation

**Lorenz Kuhn, Yarin Gal, Sebastian Farquhar**
OATML Group, Department of Computer Science, University of Oxford
`lorenz.kuhn@cs.ox.ac.uk`

## Abstract

We introduce a method to measure uncertainty in large language models. For tasks like question answering, it is essential to know when we can trust the natural language outputs of foundation models. We show that measuring uncertainty in natural language is challenging because of 'semantic equivalence'—different sentences can mean the same thing. To overcome these challenges we introduce *semantic entropy*—an entropy which incorporates linguistic invariances created by shared meanings. Our method is unsupervised, uses only a single model, and requires no modifications to 'off-the-shelf' language models. In comprehensive ablation studies we show that the semantic entropy is more predictive of model accuracy on question answering data sets than comparable baselines.

## 1 Introduction

Despite progress in natural language generation (NLG) tasks like question answering or abstractive summarisation (Brown et al., 2020; Hoffmann et al., 2022; Chowdhery et al., 2022), there is little understanding of *uncertainty* in foundation models. Without measures of uncertainty in transformer-based systems it is hard to use generated language as a reliable source of information. Reliable measures of uncertainty have been identified as a key problem in building safer AI systems (Amodei et al., 2016; Hendrycks et al., 2022).

Unfortunately, uncertainty in free-form NLG faces unique challenges. This limits how much we can learn from uncertainty estimation techniques in other applications of deep learning (Gal et al., 2016; Lakshminarayanan et al., 2017; Ovadia et al., 2019) which focuses especially on image classification (Kendall & Gal, 2017) or regression in low-dimensional data spaces (Kuleshov et al., 2018).

The key challenges come from the importance in language of *meanings* and *form*. This corresponds to what linguists and philosophers call the *semantic content* of a sentence and its *syntactic* or *lexical* form. Foundation models output *token*-likelihoods—representing lexical confidence. But for almost all applications we care about meanings! For example, a model which is uncertain about whether to generate "France's capital is Paris" or "Paris is France's capital" is not uncertain in any important sense. Yet, at a token-level the model is uncertain between two *forms* of the same *meaning*. Existing unsupervised methods (e.g., Malinin & Gales (2020)) ignore this distinction.

To address semantic equivalence, we estimate semantic likelihoods—probabilities attached to *meanings* of text rather than standard sequence-likelihoods. We introduce an algorithm for clustering sequences that mean the same thing based on the principle that two sentences mean the same thing if you can infer each from the other. We then use these semantic-likelihoods to estimate semantic uncertainty—uncertainty over different meanings. In particular, we compute the entropy of the probability distribution over meanings. Adjusting for semantic equivalence in this way offers better uncertainty estimation than standard entropy and also greatly improves over methods for model self-evaluation (Kadavath et al., 2022). In addition, semantic entropy scales better with model size and makes better use of increasing numbers of samples than baselines.

We further analyse major challenges for measuring uncertainty in NLG. We show empirically how sampling a set of model answers to estimate entropies in NLG must balance sample accuracy and diversity, which significantly strengthens the baselines we compare against relative to prior imple-

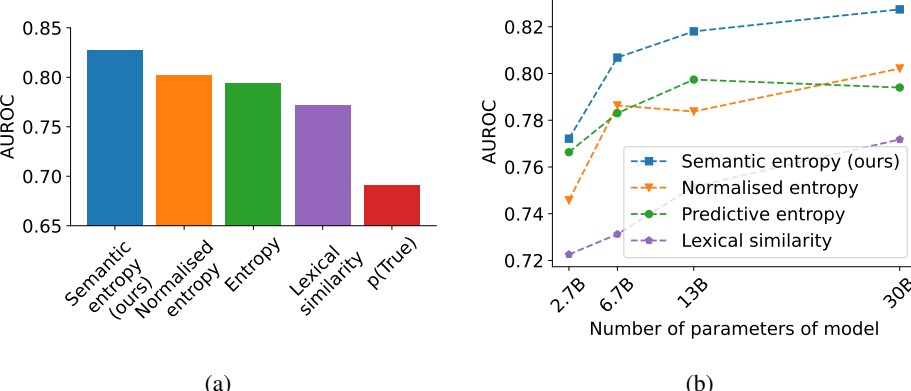

(a)                                                            (b)

Figure 1: (a) Our semantic entropy (blue) predicts model accuracy better than baselines on the free-form question answering data set TriviaQA (30B parameter OPT model). Normalised entropy reimplements single-model variant of Malinin & Gales (2020), lexical similarity measures the average Rouge-L in a sampled set of answers for a given question analogously to Fomicheva et al. (2020), entropy and $p(\text{True})$ reimplement Kadavath et al. (2022). (b) Our method's outperformance increases with model size while also doing well for smaller models.

mentations. We also examine the situational heuristic of length-normalising predictive entropies. Our main contributions are thus as follows:

- We explain why uncertainty in free-form NLG is different from other settings (Section 3).
- We introduce *semantic entropy*—a novel entropy-based uncertainty measure which uses our algorithm for marginalising over semantically-equivalent samples (Section 4) and show that it outperforms comparable baselines in extensive ablations with both open- and closed-book free-form question answering using TriviaQA and CoQA (Section 6).
- Through hyperparameter ablations we suggest how to balance the trade-off between sampling diverse and accurate generations for our method as well as baselines (Section 6.2) and show that far fewer samples are needed for effective uncertainty than prior work presumes.

We focus on free-form question answering (QA) because it is a difficult and important use of NLG with high-stakes applications. At the same time, it is easier to establish a ground truth without expensive human evaluation than more nebulous tasks like summarisation.

Ultimately, we show that semantic entropy is an effective unsupervised way to estimate uncertainty in NLG. As an unsupervised method, it requires no further training or data-gathering, unlike supervised methods including Lin et al. (2022a); Kadavath et al. (2022). Semantic entropy is designed to work with existing foundation and large language models with no modifications 'out-of-the-box'. Our experiments use OPT (Zhang et al., 2022) but semantic entropy works with any similar model.

## 2 BACKGROUND ON UNCERTAINTY ESTIMATION

Our method draws inspiration from probabilistic tools for uncertainty estimation, which have been extensively employed in settings like deep image classification (Gal et al., 2016). Although these methods are often used in Bayesian models, we emphasise that our method does not require any special training or architectural modifications and is not limited to Bayesian settings.

The total uncertainty of a prediction can be understood as the predictive entropy of the output distribution. This measures the information one has about the output given the input. This entropy is highest when the output is minimally informative—predicting the same probability for all possible outcomes. The predictive entropy for a point $x$ is the conditional entropy of the output random variable $Y$ with realisation $y$ given $x$

$$PE(x) = H(Y \mid x) = -\int p(y \mid x) \ln p(y \mid x) dy \qquad (1)$$

One can further distinguish aleatoric uncertainty—uncertainty in the underlying data distribution—and epistemic uncertainty—resulting from missing information (Kendall & Gal, 2017). Epistemic

uncertainty, measured using a mutual information, can be useful but is hard to estimate, especially for very large models, requiring special methods and computational expense. Instead of estimating the epistemic uncertainty based on the model variance, the epistemic uncertainty can also be predicted directly using a second model (see e.g. Jain et al. (2021)). We do not use mutual information in this work, because our focus is on existing foundation models 'off-the-shelf'. Similarly, while, e.g., Malinin & Gales (2020) use ensembles of models to estimate the integral in Eq. (1) we use samples from a single model's output distribution. Prior networks (Malinin & Gales, 2018; Malinin et al., 2020) estimate model uncertainty by emulating an ensemble with a single model. This could be important for NLG because of large model sizes.

For sequence-prediction tasks like NLG, the probability of the entire sequence, $\mathbf{s}$, is the product of the conditional probabilities of new tokens given past tokens, whose resulting log-probability is $\log p(\mathbf{s} \mid x) = \sum_i \log p(s_i \mid \mathbf{s}_{<i})$, where $s_i$ is the $i$'th output token and $\mathbf{s}_{<i}$ denotes the set of previous tokens. Sometimes, instead of the entropy of these probabilities, the geometric mean token-probability is used instead (Malinin & Gales, 2020) becoming an arithmetic mean log-probability $\frac{1}{N} \sum_i^N \log p(s_i \mid \mathbf{s}_{<i})$. Despite empirical success Murray & Chiang (2018), so far this has little theoretical justification.

**Direct application of language models to uncertainty.** In contrast to our approach using probabilistic methods, recent work has sought to use the generating language model itself to estimate its own uncertainty. For example, Lin et al. (2022a) finetune language models to verbally describe their confidence. Meanwhile, Kadavath et al. (2022) sample multiple generations and return the completion to an NLG prompt asking if a proposed answer is true (further detail in Appendix B.5). Both Lin et al. (2022a) and Kadavath et al. (2022) also propose ways to finetune predictors on the embeddings of generating models to predict models uncertainty. While promising, these approaches need task-specific labels, additional training, and seem to be unreliable out-of-distribution (as shown in Figures 13 and 14 in Kadavath et al. (2022)).

## 3 CHALLENGES IN UNCERTAINTY ESTIMATION FOR NLG

Approaches to NLG uncertainty might treat the language model as a black-box (e.g., asking it if its answer is correct) or alternatively focus on the probabilistic model without accounting for the special characteristics of language (e.g., measuring predictive entropy).

Our unsupervised approach instead uses the powerful tools of probabilistic modelling, but also recognises the unique challenges posed by free-form NLG. In this section, we critically analyse the probabilistic interpretation of language models in order to ground both our method and future exploration of the field.

### 3.1 SEMANTIC EQUIVALENCE IN LANGUAGE OUTPUTS

Most machine learning problems have mutually exclusive outputs. An image in class 17 is not class 29 as well; a regression output of 23.1 is not anything else; an RL agent going left does not go right. In contrast, for free-form text generation an output usually means the same thing as many other outputs. For example, "The capital of France is Paris" means the same thing as "France's capital is Paris". Linguists and philosophers distinguish text's meaning—its semantic content—from its syntactic and lexical form. The syntax is the grammatical structure while its lexical form is the specific words used. Lexical equivalence entails the other two, but not the reverse.

We almost always care about the semantic content of a sentence. For decision-problems relying on NLG, *meaning* is usually an invariance in output-space which is not present in the model specification. This is true for question answering, summarisation, artificial assistants. Meanings are especially important for trustworthiness: a system can be reliable even with many different ways to say the same thing but answering with inconsistent meanings shows poor reliability.

We can formalize semantic equivalence mathematically. Let the space of tokens in a language be $\mathcal{T}$. The space of all possible sequences of tokens of length $N$ is then $\mathcal{S}_N \equiv \mathcal{T}^N$. For some sentence $\mathbf{s} \in \mathcal{S}_N$, a sequence of tokens $s_i \in \mathcal{T}$ there is an associated meaning.[1]

Let us introduce a placeholder *semantic equivalence relation*, $E(\cdot, \cdot)$, which holds of any two sentences that mean the same thing—we operationalise this in Section 4. Recall that an equivalence

---

[1]Theories of meaning are contested (Speaks, 2021). However, for specific models and deployment contexts many considerations can be set aside. Care should be taken comparing very different models and contexts.

Table 1: Answers to the question "What is the capital of France?" (a) When all generations from the model mean different things, semantic clustering has no effect—the entropy and semantic entropy are identical. (b) When some of the answers are semantically equivalent ("Paris" and "It's Paris") the semantic entropy does a better job of capturing the actually low uncertainty.

| (a) Scenario 1: No semantic equivalence | | | (b) Scenario 2: Some semantic equivalence | | |
|---|---|---|---|---|---|
| Answer $\mathbf{s}$ | Likelihood $p(\mathbf{s} \mid x)$ | Semantic likelihood $\sum_{\mathbf{s} \in c} p(\mathbf{s} \mid x)$ | Answer $\mathbf{s}$ | Likelihood $p(\mathbf{s} \mid x)$ | Semantic likelihood $\sum_{\mathbf{s} \in c} p(\mathbf{s} \mid x)$ |
| Paris | 0.5 | 0.5 | **Paris** | 0.5 } | 0.9 |
| Rome | 0.4 | 0.4 | **It's Paris** | 0.4 } | |
| London | 0.1 | 0.1 | London | 0.1 | 0.1 |
| Entropy | 0.31 | 0.31 | Entropy | 0.31 | 0.16 |

relation is any reflexive, symmetric, and transitive relation, and that any equivalence relation on a set corresponds to a set of equivalence classes. Each semantic equivalence class corresponds to one possible meaning that our text can have. That is, for the space of semantic equivalence classes $\mathcal{C}$ the sentences in the set $c \in \mathcal{C}$ all share a meaning such that $\forall s, s' \in c : E(s, s')$.

Ordinarily, large language models produce conditional distributions over tokens and their resulting sequences. That is, the probability of the sequence conditioned on the context comes from conditional token probabilities $p(\mathbf{s} \mid x) = \prod_i p(s_i \mid s_{<i}, x)$. Instead, we focus on the probability of the model generating any sequence that shares some meaning. This can be written as

$$p(c \mid x) = \sum_{\mathbf{s} \in c} p(\mathbf{s} \mid x) = \sum_{\mathbf{s} \in c} \prod_i p(s_i \mid s_{<i}, x). \tag{2}$$

Formally, this treats the output random variable whose event-space is $\mathcal{C}$, a sub-$\sigma$-algebra of the standard event-space $\mathcal{S}$.

### 3.2 Sampling the extremely high-dimensional language-space

Recall from Eq. (1) that estimating predictive entropy requires taking an expectation in output-space. However, the output-space of natural language has $\mathcal{O}(|\mathcal{T}|^N)$ dimensions. Moreover, while we can sample from our autoregressive token-model, we lack a normalized probability density function over sentences. The expectation must be approximated by Monte Carlo integration—sampling a finite set of sentences from the output distribution and averaging their likelihoods to compute the entropy. For entropies the average is dominated by low-probability sentences (whose logs are large and negative) making Monte Carlo integration difficult (Mackay, 2003).

### 3.3 Variable length generations

Sentences of natural language have different lengths. As is widely noted (Murray & Chiang, 2018) and especially in the context of NLG uncertainty by Malinin & Gales (2020), in expectation longer sequences have lower joint likelihoods because of the conditional independence of the token probabilities. The joint likelihood of a sequence of length $N$ shrinks exponentially in $N$. Its negative log-probability therefore grows linearly in $N$, so longer sentences tend to contribute more to entropy.

We therefore interpret length-normalising the log-probabilities when estimating the entropy as asserting that the expected uncertainty of generations is independent of sentence length. Sometimes, this is approximately valid. Other times, longer sentences may well be usually more uncertain (e.g., when the goal is to exactly match a typically short reference answer, such as for TriviaQA). In these cases, the advantages of length-normalisation become less clear-cut, as we show empirically in Section 6.1. This offers some guidance *a priori* on cases when length-normalisation is appropriate.

## 4 Semantic Uncertainty

We have introduced the idea that uncertainty over *meanings* is more important for most situations than uncertainty over the exact tokens used to express those meanings. Our method examines uncertainty in meaning-space—the entropy of the random variable representing the output distribution in the semantic event-space. This is in contrast to entropy in the usual token event-space. To do this we introduce a novel algorithm for estimating the semantic equivalence relation as well as a novel uncertainty estimation algorithm for semantic entropy. At a high level this involves three steps:

1. **Generation:** Sample $M$ sequences $\{s^{(1)}, \ldots, s^{(M)}\}$ from the predictive distribution of a large language model given a context $x$.

2. **Clustering:** Cluster the sequences which mean the same thing using our bi-directional entailment algorithm.

3. **Entropy estimation:** Approximate semantic entropy by summing probabilities that share a meaning following Eq. (2) and compute resulting entropy. This is illustrated in Table 1.

**Step 1: Generating a set of answers from the model**

First we sample $M$ sequences $\{s^{(1)}, \ldots, s^{(M)}\}$ which we will use later to estimate the uncertainty. These sequences must be sampled according to the distribution $p(\mathbf{s} \mid x)$. In this paper, we sample these sequences only from a *single* model using either multinomial sampling or multinomial beam sampling. We show in Section 6.2, that the choice of sampling temperature and sampling method can have a significant impact on the performance of both our method and the baselines. Unlike Malinin & Gales (2020), we do not use an ensemble of models. Ensembling would probably improve performance, but the cost of training multiple independent foundation models is often prohibitive.

**Step 2: Clustering by semantic equivalence**

In Section 3.1, we formalised semantic equivalence by introducing the semantic equivalence relation, $E(\cdot, \cdot)$, which induces semantic equivalence classes which are sets of sequences that share a meaning. Recall that the equivalence class $c$ is a set of sequences $\mathbf{s}$ such that $\forall s, s' \in c : E(s, s')$. We operationalise $E(\cdot, \cdot)$ using the idea of bi-directional entailment. A sequence, $\mathbf{s}$, means the same thing as a second sequence, $\mathbf{s}'$, if and only if they entail (i.e. logically imply) each other. E.g., "The capital of France is Paris." entails "Paris is the capital of France." because they mean the same thing.

Importantly, we require that the sequences mean the same thing with respect to the context—key meaning is sometimes contained within the context. For example, "Paris." does not entail "The capital of France is Paris." because "Paris." is not a declarative sentence without context. But within the context of the question, the one-word answer does entail the fuller answer.

In general, any natural language inference classification system (NLI) can be used for our bidirectional entailment clustering algorithm. In our case, we use a Deberta-large model (He et al., 2020a) that is fine-tuned on the NLI data set MNLI (Williams et al., 2017). For each pair of sequences in our set of samples, $\mathbf{s}$ and $\mathbf{s}'$, we detect whether it is possible to infer the concatenation of the context and $\mathbf{s}$ from the concatenation of the context and $\mathbf{s}'$ and vice versa. To do this we concatenate each of the two question/answer pairs, and then concatenate them both together separated by a special token. The Deberta model then classifies this sequence into one of: `entailment`, `neutral`, `contradiction`. We compute both directions, and the algorithm returns `equivalent` if and only if both directions were `entailment`. Algorithm pseudocode is provided in Appendix A.2.

Because this component is novel, we confirm in Appendix B.2 that the bidirectional entailment classifier works by manually labelling 300 generations for semantic equivalence, finding an accuracy of 92.7% on TriviaQA and 95.5% on CoQA.

**Computational cost.** The bidirectional equivalence algorithm is combinatorially complex in $M$, it requires $\binom{M}{2}$-many comparisons in the worst-case. In practice, however, the computational cost is small compared to the cost of generating sequences. First, as we show in Section 6.2, $M < 20$ is often sufficient for good uncertainty. Second, because the Deberta-large model is so much smaller than the main language model (1.5B parameters), each pair comparison is much faster than generating even one token from the main model. Third, because semantic equivalence is transitive we only need to compare one member of each equivalence class to remaining sequences (see Algorithm 1). Additionally, the number of semantic clusters in our tasks is empirically quite low, see Table 2.

**Step 3: Computing the semantic entropy**

Having determined the clusters of generated sequences that mean the same thing, we add their likelihoods following Eq. (2) as a way of determining the likelihood of each meaning, rather than each sequence. We then compute the semantic entropy (SE) as the entropy over the meaning-distribution

$$SE(x) = -\sum_c p(c \mid x) \log p(c \mid x) = -\sum_c \left( \left( \sum_{\mathbf{s} \in c} p(\mathbf{s} \mid x) \right) \log \left[ \sum_{\mathbf{s} \in c} p(\mathbf{s} \mid x) \right] \right). \quad (3)$$

We do not have access to every possible meaning-class $c$. Instead, we can only sample $c$ from the sequence-generating distribution induced by the model. To handle this, we estimate the expectation in Eq. (3) using Monte Carlo integration over the semantic equivalence classes $C$ from Algorithm 1

$$SE(x) \approx -|C|^{-1} \sum_{i=1}^{|C|} \log p(C_i \mid x).$$ (4)

This is an unbiased estimator of the entropy in Eq. (3). In addition, in some cases we use length-normalisation as described in Section 3.3.

## 4.1 How the semantic entropy addresses the challenges of NLG

The main inspiration of semantic entropy is to address the semantic invariance of natural language head-on by converting the problem of uncertainty estimation into meaning-space. In addition, semantic entropy goes some way towards addressing unequal token importance. Generations whose meanings are the same but differ on unimportant tokens will be added together, which we expect will reduce the effect of the likelihoods of unimportant tokens although we do not demonstrate this empirically. However, this challenge is only partially addressed: semantic entropy will still pay too much attention to non-keyword likelihoods. This is one area where supervised language-model-based uncertainty tools (Lin et al., 2022a; Kadavath et al., 2022) might enable future improvements that handle this challenge better. We address the challenges of sampling and variable-length generation using specific details of our sampling procedure in Section 4.

## 5 Related Work

Prior work on uncertainty in foundation models for NLP has largely focused on the calibration of *classifiers* (Jiang et al., 2021; Desai & Durrett, 2020) and text regressors (Glushkova et al., 2021; Wang et al., 2022). These settings, are analogous to classification or regression settings in other modalities like vision, and conventional uncertainty measures like MC dropout or Deep Ensembles can be applied without modification (see Section 2 for a discussion of uncertainty in deep learning in general). As we argue in Section 3, generative natural language poses important further challenges. Jiang et al. (2021) do examine calibration in generative question answering and find only a weak correlation between the log-likelihood models assign to their answer and the answer's correctness. In Section 6 we explain however why semantic equivalence in natural language makes calibration a problematic evaluation for generative language models. Reliable uncertainty can be useful on downstream tasks such as graph semantic parsing (Lin et al., 2022b).

Some research has addressed uncertainty or calibration in NLG either by prompting the models to evaluate their own generations or by fine-tuning the generating model to predict its uncertainty (Mielke et al., 2020; Lin et al., 2022a; Kadavath et al., 2022). These methods need further training and supervision. Because they need additional training and supervision, they are hard to reproduce, expensive to create, and have been shown to be sensitive to distribution shift. For example, we were unable to implement one proposal by Kadavath et al. (2022) to train a language model to directly predict confidence due to hardware limitations. Our unsupervised method which uses models 'off-the-shelf' avoids these limitations.

Many of the issues that make probabilistic uncertainty estimation in NLG difficult also make automatic evaluation of NLG difficult. Ott et al. (2018), for instance, study how the performance of machine translation models suffers because one sentence can be translated in multiple ways. Similarly, Sai et al. (2022) discuss how paraphrase detection can be used to evaluate NLG and other related methods might transfer to uncertainty estimation.

Automatic paraphrase identification can be based on comparing lexical features of two given sequences (Fernando & Stevenson, 2008; Issa et al., 2018) or on measuring the similarity between the embeddings of the two sequences (Yu et al., 2014; Socher et al., 2011). Recently, however, SotA paraphrase identification approaches have primarily used BERT-based models to classify pairs of sequences into the classes `paraphrases` and `not paraphrases` (He et al., 2020b; Tay et al., 2021). The idea of formalising semantic equivalence via textual entailment has a long history in linguistics (Culicover, 1968) and NLP (Padó et al., 2009; Androutsopoulos & Malakasiotis, 2010). Transformer-based paraphrase detection models such as EFL (Wang et al., 2021) achieve SotA performance on paraphrase detection benchmarks such as Quora Question Pairs Wang et al. (2017).

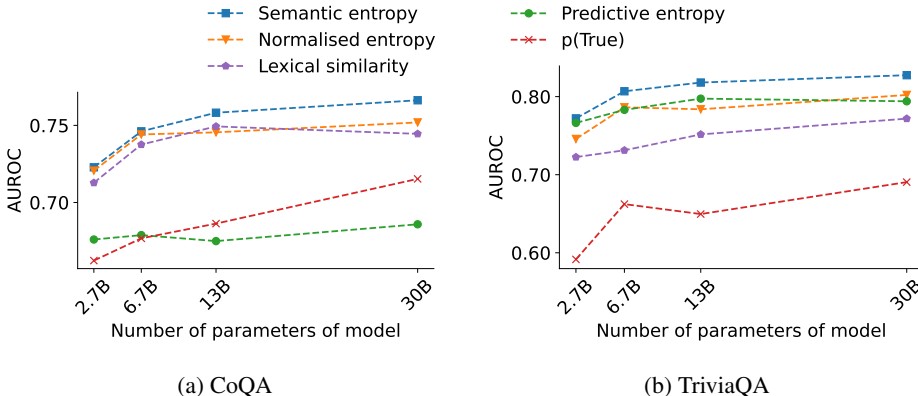

(a) CoQA
(b) TriviaQA

Figure 2: (a) On CoQA open-book question answering semantic entropy demonstrates better uncertainty than ordinary predictive entropy with and without normalisation at larger model sizes. It also performs significantly better than $p$(True). (b) TriviaQA shows similar results. Identical to Fig. 1b with the addition of $p$(True), which was previously omitted to avoid stretching the scale.

## 6 EMPIRICAL EVALUATION

We demonstrate that semantic entropy is an effective way to quantify the uncertainty of NLG on free-form QA tasks. Effective uncertainty measures should offer information about how reliable the model's answers are—that is, very *uncertain* generations should be less likely to be *correct*.

**Performance evaluation.** Following prior work (e.g. Filos et al. (2019)), we evaluate uncertainty by treating uncertainty estimation as the problem of predicting whether to rely on a model generation for a given context—whether to trust an answer to a question. The area under the receiver operator characteristic curve (AUROC) metric is equivalent to the probability that a randomly chosen correct answer has a higher uncertainty score than a randomly chosen incorrect answer. Higher scores are better, with perfect uncertainty scoring 1 while a random uncertainty measure would score 0.5.

The AUROC is a better measure of uncertainty for *free-form* question answering and NLG than calibration measures like the Brier score, which are often used in classification or for multiple choice QA. This is because the language model outputs a likelihood for a given token-sequence, but not for an entire meaning. In order to estimate the Brier score, we would need to estimate the entire probability mass assigned to any possible way of saying the correct answer. This is intractable for free form text where we do not have access to probabilities about meanings. In contrast, we can estimate the entropy because it is structured as an expected information, which makes Monte Carlo integration suitable.

**Model.** We use the GPT-like OPT models (Zhang et al., 2022). We vary the size of the model between 2.7B, 6.7B, 13B and 30B parameters, while our headline results are all reported using the largest computationally feasible model, with 30B parameters. In all cases we use only a single unmodified model. There is no ensembling and no stochastic or Bayesian modification. We chose this because in most cases cutting-edge foundation models are not available as ensembles and are too large to efficiently perform approximate Bayesian inference with. We do not fine-tune these models on TriviaQA or CoQA but use them in their pre-trained form.

**Datasets.** We use CoQA Reddy et al. (2019) as an open-book conversational question answering problem (in which the model answers a question using a supporting paragraph). We use the development split (∼8000 questions). We also use TriviaQA (Joshi et al., 2017) as a closed-book QA problem (in which the model must answer a question without access to a supporting paragraph). We use a subset of 8000 questions of the training split to match the size of CoQA.

We evaluate correctness of our model's generations on the underlying dataset using the a fuzzy matching criterion: $\mathcal{L}(\mathbf{s}, \mathbf{s}') = \mathbf{1}_{RougeL(\mathbf{s}, \mathbf{s}') > 0.3}$. That is, we consider an answer $\mathbf{s}$ to be correct if its Rouge-L (Lin & Och, 2004) — a measure of the longest common subsequence — with regards to the reference answer is larger than 0.3. In Appendix B.3 we study other objective functions such as exact matching and Rouge-1 and find our method to be robust to these choices.

**Baselines.** We compare our method against predictive entropy, length-normalised predictive entropy (Malinin & Gales, 2020), $p$(True) (Kadavath et al., 2022), and lexical similarity (similar to (Fomicheva et al., 2020)). **Predictive entropy** is a widely used measure of uncertainty in other

Table 2: Incorrectly answered questions have more semantically distinct answers than correct ones. On its own, this count is a reasonable uncertainty measure, though semantic entropy is better.

| Dataset | Average # of semantically distinct answers | | AUROC | |
|---------|------------------|-------------------|------------------|--------------------|
|         | Correctly answered | Incorrectly answered | Semantic entropy | # distinct answers |
| CoQA    | 1.27             | 1.77              | 0.77             | 0.66               |
| TriviaQA | 1.89            | 3.89              | 0.83             | 0.79               |

domains, and has been used as a baseline without length-normalisation in, e.g., Kadavath et al. (2022). Here, the score is just the predictive entropy of the output distribution as described in Eq. (1). **Length-normalised predictive entropy** divides the joint log-probability of each sequence by the length of the sequence, as proposed by Malinin & Gales (2020) in the case of NLG uncertainty and further discussed in Section 3.3. Note that unlike Malinin & Gales (2020), we use only a single model, not an ensemble, and use multinomial sampling as we do for all other methods. $p(\textbf{True})$ proposed by (Kadavath et al., 2022) as a way to estimate the probability that a model's generation is correct by 'asking' the model if its answer is correct. They propose sampling $M$ answers and constructing a new natural language question using these possible answers as context before asking whether the proposed answer is correct and measuring the probability of the completion being `True`. An example of the format is provided in Appendix B. Note that our implementation of this uses OPT models with up to 30B parameters, while Kadavath et al. (2022) use a proprietary 52B parameter model. We are also limited computationally to 10-shot prompting while the original paper uses 20-shot prompting. **Lexical similarity** uses the average similarity of the answers in the answer set $\mathbb{A}$: $\frac{1}{C} \sum_{i=1}^{|\mathbb{A}|} \sum_{j=1}^{|\mathbb{A}|} \text{sim}\,(s_i, s_j)$, where $C = |\mathbb{A}| * (|\mathbb{A}| - 1)/2$, and $sim$ is Rouge-L. Additionally, we evaluate a margin-probability baseline (Lin et al., 2022b) in Appendix B.6, and study why it is not very predictive of model accuracy in this setting. All code and data used in our experiments are available at `https://github.com/lorenzkuhn/semantic_uncertainty`.

## 6.1 SEMANTIC ENTROPY UNCERTAINTY

For both TriviaQA and CoQA, semantic entropy improves over baselines in predicting whether a model's answer to a question is correct. For TriviaQA, using the largest model we show in Fig. 1a we show that semantic entropy has a significantly higher AUROC than entropy in sequence-probability-space with and without length-normalisation, as well as the lexical similarity baseline. At the same time, it performs dramatically better than $p(\text{True})$. Similarly, we find in Fig. 1b that our method outperforms more for larger model sizes and continues to steadily improve as the model size increases, with the performance of the $p(\text{True})$ baseline added in Fig. 2b (not shown in the opening figure for visual clarity). For CoQA, in Fig. 2a we show that semantic entropy predicts model correctness significantly better than the baselines at larger model sizes.

The ground truth answers for TriviaQA are generally single words or very short phrases, while CoQA contains both longer and shorter ground truth answers. This is why performing length-normalisation has a large effect for CoQA but no effect for TriviaQA (compare Fig. 2a and Fig. 2b). TriviaQA is also a more challenging dataset: accuracy of 50.6% against 82.3% for CoQA.

We can better understand the mechanism of action for semantic entropy by examining the difference between incorrect and correct answers generated by the model. In Table 2 we show that the average number of semantically distinct clusters of answers ($|C|$) from the 30B parameter model is significantly greater for incorrectly answered questions than correctly answered ones. This fits our predictions, which is that the model is more likely to generate incorrect answers when it is uncertain about the most likely generation. There are 10 answers generated per question, so there is substantial overlap in meaning. We also show that simply using the number of semantically distinct answers as an uncertainty measure on its own performs reasonably well. Semantic entropy has a higher AUROC than the number of distinct answers, especially for CoQA whose difficulty causes greater spread in predicted probabilities between possible answers.

Finally, we can see that much of the performance gain comes from making better use of more samples. In Fig. 3a we show that for both CoQA (top) and TriviaQA (bottom) the gap between semantic entropy and length-normalised entropy widens as the number of samples increases.

## 6.2 HYPERPARAMETERS FOR EFFECTIVE SAMPLING

Adjusting the temperature used for multinomial sampling has two competing effects on the generated sequences produced by the model. Increasing the temperature increases the diversity of samples

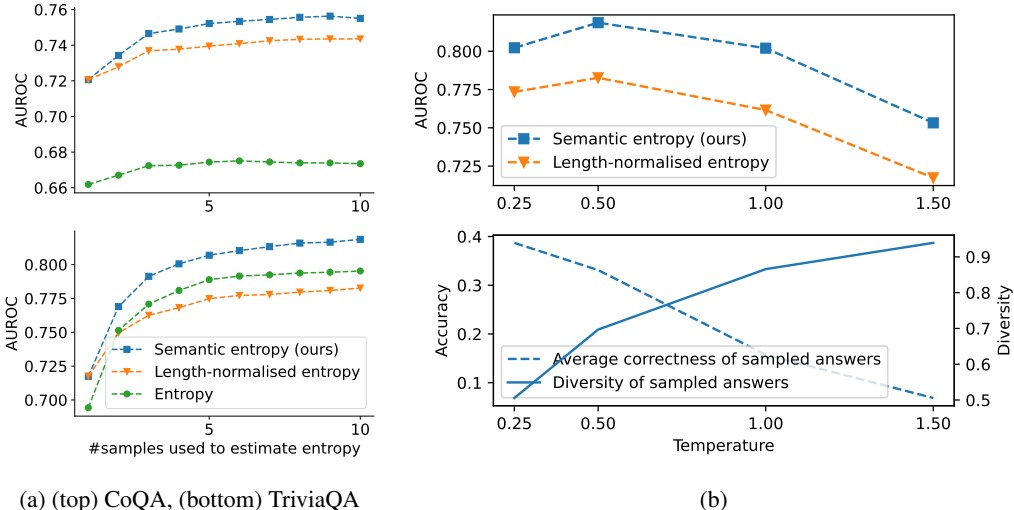

(a) (top) CoQA, (bottom) TriviaQA    (b)

Figure 3: (a) Semantic entropy makes better use of additional samples because it handles duplication better, the performance gap therefore continues to improve. (b) (bottom) Higher temperatures result in more diversity but less accurate generations. (top) The best performing uncertainty comes from an intermediate temperature that balances these two forces. Results on TriviaQA.

(Fig. 3b, bottom figure, solid line). One would expect more diverse generations to cover the space of possible meanings more fully. Here we measure the diversity using the average overlap of the longest sub-sequence among sampled answers $(1 - \binom{M}{2}^{-1} \sum_{\mathbf{s} \neq \mathbf{s}' \in C} \text{Rouge-L}(\mathbf{s}, \mathbf{s}'))$. At the same time, reducing the temperature improves the average correctness of the answer (Fig. 3b, bottom figure, dashed line). Typically, more accurate models are also better at estimating uncertainty.

In fact, we find that these two effects compete and the highest AUROC for semantic entropy and length-normalised entropy is optimised by an intermediate temperature of 0.5 (Fig. 3b, top figure). A lower temperature would improve accuracy, while a higher temperature would improve diversity. A similar figure for CoQA can be found in Appendix B. Note that prior work using predictive entropy as a baseline uses a temperature of 1.0 (Kadavath et al., 2022), which our evaluation suggests would significantly weaken the baseline relative to our implementation.

# 7 DISCUSSION

Many natural language problems display a crucial invariance: sequences of distinct tokens *mean* the same thing. Addressing this directly, we introduce semantic entropy—the entropy of the distribution over meanings rather than sequences—and show that this is more predictive of model accuracy on QA than strong baselines. Our unsupervised approach using 'out-of-the-box' models improves reproducibility and is easier to deploy. Unsupervised uncertainty may also help address the observation raised in prior work that supervised uncertainty measures struggle with distribution shift.

For semantic entropy, we introduce a novel bidirectional entailment clustering algorithm which uses a smaller natural language inference model. Our method therefore represents a middle ground between fully probabilistic methods and methods that use language models to exploit aspects of natural language that are not transparently present in the model activations. We believe that this sort of joint approach is more promising than relying on either perspective on its own, especially as language models continue to improve. This will become more important in cases where language models are capable of deception, something which our method does not protect against, rather than merely being uncertain between many possible meaningful options. By combining model internals with model prediction one can hope to stay a step ahead of model capabilities.

We focus on question answering because this is a particularly important free-form NLG problem with relatively clear ground truths. In the future, however, we hope our work on semantic equivalence can pave the way towards progress in settings like summarisation where correctness requires more human evaluation although additional progress on paraphrase identification in these settings is likely required first. Semantic likelihoods could also be extended to other tools for probabilistic uncertainty like mutual information, potentially offering new strategies for NLG uncertainty.

ACKNOWLEDGMENTS

We are grateful to Geoffrey Irving, Kuba Perlin, Laura Rimell, and Miles Turpin for their advice and feedback on earlier drafts of this paper. We are also grateful to the members of the OATML group for helpful discussions about this project.

ETHICS STATEMENT

Our aim is to work towards safer AI systems by enabling users to understand the confidence and reliability of language model generations. In principle, this could help mitigate many of the potential harms of NLG from foundation models such as generating false and harmful information in response to genuine questions about important topics like medical questions. However, this potential benefit comes with the risk that systematic mistakes in the assessment of uncertainty or its communication could cause unfounded and misplaced confidence. While this paper represents research progress in identifying new considerations and methods for uncertainty quantification in NLG, before deployment we advise that practitioners conduct extensive evaluations specific to the deployment context to make sure that uncertainty is communicated in a way that empowers users and is not misleading or confusing.

REPRODUCIBILITY STATEMENT

Because of the computational cost of experimentation with foundation models, most of the relatively small amount of existing research into NLG uncertainty relies on proprietary models, finetuning of expensive models, and human evaluation. These factors put this kind of research out of reach for many academic groups. Our work takes advantage of the recently released, publicly available OPT models, and builds on this to provide an uncertainty quantification pipeline for NLG that uses entirely open source tools. Meanwhile our method requires no finetuning or training of foundation models and can work with 'off-the-shelf' existing models. We hope that this can facilitate more research on these important topics in the academic community as well as making our methods easier to replicate. We make all of our code, as well as the hand-labelled semantic equivalence dataset drawn from TriviaQA and CoQA, available under an MIT license.

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

Table 3: Illustration of semantic, syntactic, and lexical equivalence. Work with foundation models implicitly focuses on *lexical* equivalence, which entails the others, but we usually care about *semantic* equivalence.

| | | Equivalence | | |
|---|---|---|---|---|
| Sentence A | Sentence B | Lexical | Syntactic | Semantic |
| Paris is the capital of France. | Paris is the capital of France. | ✓ | ✓ | ✓ |
| | Berlin is the capital of France. | | ✓ | |
| | France's capital is Paris. | | | ✓ |

## A  FURTHER DETAILS ON SEMANTIC ENTROPY

### A.1  FURTHER DISCUSSION OF SEMANTIC EQUIVALENCE

We illustrate the distinction between different kinds of equivalence in Table 3. Lexically equivalent sequences use exactly the same symbols. They are always also semantically and syntactically equivalent (in a given context). Syntactically equivalent sentences have the same grammatical form. But they can have different meanings (not semantically equivalent) and can use different symbols (not lexically equivalent). Semantically equivalent sentences mean the same thing, but they might have different grammatical form (not syntactically equivalent) or symbols (not lexically equivalent). Two sentences can also be both syntactically and semantically equivalent but not lexically equivalent if they match up to a synonym.

**Soft equivalence and transitivity.** Formally, semantic equivalence is transitive. That is, if $E(\mathbf{s}, \mathbf{s}')$ and $E(\mathbf{s}', \mathbf{s}'')$ then it follows that $E(\mathbf{s}, \mathbf{s}'')$. However, the implementation of our bidirectional equivalence algorithm permits some classification errors and it is slightly 'soft'—it will sometimes return `equivalent` for pairs that are not *quite* equivalent. As a result, it is not strictly true that our equivalence relation is transitive, and therefore not strictly true that there is a unique set of equivalence classes. For example, the clusters might depend on the order in which the comparisons are made. In practice, however, we find that this does not pose a noticeable problem—usually, inspecting the outputs shows that the equivalence appears clear cut. However, we acknowledge this potential issue as an area for improvement in future clustering algorithms.

**Unequal token importance.** From the perspective of meaning, some tokens can matter more than others—key words. Naive methods like predictive entropy do distinguish between key words or unimportant tokens. Supervised uncertainty methods that make use of language models in the uncertainty evaluation can potentially take this into account better. In addition, our semantic entropy approach partly adjusts for this, as discussed in Section 4.1.

### A.2  FURTHER ALGORITHMIC DETAILS

In addition to the description of the method provided in the main body, in Algorithm 1 we provide the pseudocode for our bi-directional entailment algorithm.

### A.3  IMPACT OF SAMPLING METHOD ON QUALITY OF UNCERTAINTY ESTIMATE

In Section 4, we study the impact of the temperature hyper-parameter on the performance of the uncertainty measures. Here, we show a variant of Fig. 3b for the CoQA dataset showing an almost identical pattern. Like TriviaQA, the optimal temperature is 0.5 despite a significantly harder problem with lower accuracy, suggesting that this choice hyperparameter may generalize well. Unlike TriviaQA, normalised entropy outperforms semantic entropy at high temperatures.

Beyond the temperature, there are a number of other design choices to be made when sampling: the sampling method and hyper-parameters such as `top-p` and `top-k`. Our contribution in this paper is to show the importance of these choices for uncertainty estimation which has been overlooked previously, and study the temperature in particular. While we leave the detailed study of these hyperparameters to future work, we do compare our default multinomial sampling method, to multinomial beam search sampling which focuses more on high-likelihood regions of the output space.

---

**Algorithm 1** Bidirectional Entailment Clustering

---

**Require:** context $x$, set of seqs. $\{\mathbf{s}^{(2)}, \ldots, \mathbf{s}^{(M)}\}$, NLI classifier $\mathcal{M}$, set of meanings $C = \{\{\mathbf{s}^{(1)}\}\}$
  **for** $2 \leq m \leq M$ **do**
    **for** $c \in C$ **do**                                       ▷ Compare to already-processed meanings.
      $\mathbf{s}^{(c)} \leftarrow c_0$                                ▷ Use first sequence for each semantic-class.
      `left` $\leftarrow \mathcal{M}(\text{cat}(x, \mathbf{s}^{(c)}, \text{"<g/>"}, x, \mathbf{s}^{(m)}))$      ▷ Does old sequence entail new one?
      `right` $\leftarrow \mathcal{M}(\text{cat}(x, \mathbf{s}^{(m)}, \text{"<g/>"}, x, \mathbf{s}^{(c)}))$         ▷ Vice versa?
      **if** `left` is `entailment` **and** `right` is `entailment` **then**
        $c \leftarrow c \bigcup \mathbf{s}^{(m)}$                  ▷ Put into existing class.
      **end if**
    **end for**
    $C \leftarrow C \bigcup \{\mathbf{s}^{(m)}\}$                    ▷ Semantically distinct, gets own class.
  **end for**
  **return** $C$

---

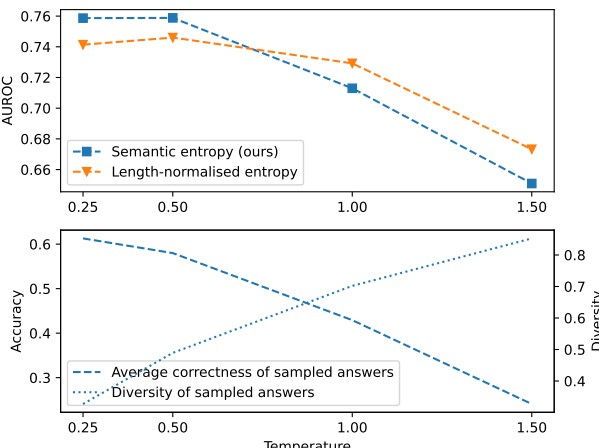

Figure 4: CoQA temperature ablation. (bottom) Similar to TriviaQA, higher temperatures mean higher diversity and lower accuracy. (top) The best performance for both methods comes at a temperature of 0.5. Unlike TriviaQA, normalised entropy outperforms semantic entropy at high temperatures.

In Table 4 we show that multinomial beam search sampling yields uncertainty measures that are less predictive of model accuracy than multinomial sampling. Beam search also generates much less diverse samples. We conjecture that multinomial beam search sampling focuses too much on the most likely sequences. The diversity of this beam search corresponds to the lowest temperature result in Fig. 4. As in the main body of the paper, we measure diversity as the average lexical overlap of the answers in the answer set. Additionally, we investigate, why the semantic entropy underperforms the length-normalised entropy at high temperatures. To that end, we manually inspect and label 100 classifications of our semantic equivalence method at T=1.5, and we find that at these temperatures, many of the generated model answers are nonsensical combinations of words from the context that is provided for the question. While the likelihood of these sequences still seems somewhat predictive of the model's accuracy, semantic clustering becomes very difficult and an unreliable signal for uncertainty estimation. At this temperature, the accuracy of the semantic equivalence methods is only 61%, whereas it is over 92% at lower temperatures (see Appendix B.2)

Note, that at low-temperatures, where one does gets plausible and well-formed model generations, semantic entropy does clearly outperform the baselines. This finding further underlines the importance of choosing appropriate sampling hyper-parameters when using entropy-based uncertainty measures in NLG.

Table 4: Multinomial beam search sampling produces sampled answers that are less diverse and thus less useful for uncertainty estimation than multinomial sampling.

| Sampling method | Semantic Entropy AUROC | Diversity of answers |
| --- | --- | --- |
| Multinomial sampling | 0.758 | 0.490 |
| Multinomial beam search sampling | 0.735 | 0.258 |

## B    EXPERIMENTAL DETAILS AND ABLATIONS

We use both the OPT models[2] and the Deberta-large model[3] via the HuggingFace transformers library which can be easily adopted for reproducibility. All of our code is open-source and relies on no proprietary models.

We use the following functions of the HuggingFace API to sample the most likely answers, and the set of answers:

- To obtain the answer which is compared to the reference answer, which determines whether the question is correctly answered, we use beam search using the `generate()` function with `num_beams = 5` and `do_sample = True`.
- To obtain the answer set for uncertainty estimation, by default we use multinomial sampling, that is `generate()` using `do_sample = True` and `num_beams = 1`. If indicated explicitly, we use beam multinomial sampling, that is `generate()` using `num_beams = 5` and `do_sample = True`.

We run all of our experiments on 80GB NVIDIA A100s.

Testing up to 20 samples per answer on the 2.7B, 6.7B and 13B CoQA experiments, we conclude that using more than 10 samples does not significantly improve the performance of the uncertainty measure, we use 10 sampled answers per question in the remaining experiments on TriviaQA. Note, that in Table 2 we compare the 30B model on CoQA and TriviaQA where in both settings we use answer sets of size 10.

We use the following prompts on CoQA and TriviaQA. We find that on CoQA, we obtain accurate model results with zero-shot prompting. While we have to use few-shot prompting to obtain accurate answers on closed-book TriviaQA. We use the following prompts for each of the settings:

**CoQA:**

```
[The provided context paragraph]
[additional question-answer pairs]
Q: [Provided question]
A:
```

where `additional question-answer pairs` are preceding turns of the conversation about the paragraph consisting of questions and reference answers.

**TriviaQA:**

For TriviaQA, we use a 10-shot prompt of the format:

```
Q: Which Oscar-nominated film had You Sexy Thing as its theme
song?  A: The Full Monty Q: Which Joan's career revived in
Whatever Happened to Baby Jane?  A: Crawford Q: Which much-loved
actor won the Best Actor Oscar for The Philadelphia Story?  A:
James Stewart (...)  Q: In which river is the Boulder Dam?  A:
```

To account for generations where the model continues the `Q:...A:...` pattern after providing an answer to the given question, we trim all generations by pattern matching for a selection of stopwords that we observe in the generations: `Q:`, `Question:`, `QUESTION:` and `questions:`.

---

[2]https://huggingface.co/docs/transformers/model_doc/opt
[3]https://huggingface.co/docs/transformers/model_doc/opt

Table 5: **Automatic evaluation of question answering is highly accurate as compared to human evaluation.** We evaluate how accurate the automatic evaluation metric. The predictions, in this settings are the automatically determined accuracy labels on our question answering task, and the ground truth are human labels for the accuracy of the provided model generation given the reference answer

| Data set | Accuracy of automatic evaluation |
|----------|----------------------------------|
| CoQA     | 0.89                             |
| TriviaQA | 0.96                             |

Table 6: TriviaQA: the exact choice of accuracy metric for the free-form QA task has little effect on the assessment of the quality of the uncertainty measure.

| Metric | AUROC | | Accuracy |
|--------|-------|--|----------|
|        | Semantic entropy | Normalised entropy | |
| Rouge-L$(y, y') > 0.3$ | 0.828 | 0.802 | 0.506 |
| Rouge-L$(y, y') > 0.5$ | 0.835 | 0.810 | 0.456 |
| Rouge-1$(y, y') > 0.5$ | 0.835 | 0.810 | 0.457 |
| Exact matching | 0.828 | 0.808 | 0.394 |

## B.1 RELIABILITY OF ACCURACY METRIC AS COMPARED TO HUMAN EVALUATION

In our experiments, we evaluate how well our uncertainty measures predict the model's accuracy when answering a given question. The choice of accuracy metric is thus a crucial component of our experimental setup. Generally, it has been shown to be difficult to develop automatic metrics for free-form generation that correlate well with human evaluations. We thus verify our choice of accuracy criterion: Rouge-L$(y, y') > 0.3$, for a given reference answer $y$ and a model generation $y'$. We manually evaluate the accuracy of 200 answers of the 30B parameter model on both COQA and on TriviaQA, and evaluate how closely the human evaluation matches the automatic evaluation. We find that on both data sets, the accuracy of the automatic labels as compared to the human labels as the ground truth is high, see Table 5.

## B.2 TESTING THE BI-DIRECTIONAL ENTAILMENT CLASSIFIER

To the best of our knowledge, this paper is the first application of the bi-directional entailment approach to identifying answers with the same meaning in question answering. Since this is a core component of our approach, we verify how accurately this approach identifies model answers with the same meaning. To this end, we manually label 300 samples for each of TriviaQA and CoQA produced by the 13B parameter model to provide a ground truth as to whether or not they mean the same thing. We find that the model achieves an accuracy of 92.7% and 95.3% respectively.

## B.3 SENSITIVITY OF RESULTS TO ACCURACY METRIC

In principle, the choice of metric to decide whether or not an answer is 'correct' might have a large effect on the assessment of our method and baselines. However, we find empirically that our results are relatively insensitive to the choice of accuracy metric.

In Table 6 we show that for TriviaQA the choice of accuracy metric for the question answering has almost no effect on the measured AUROC of the uncertainty estimation, despite making the measured accuracy of the model's generation significantly different. In particular, the exact matching requirement reduces the accuracy significantly but has little effect on the AUROCs.

For CoQA, which is an open-book QA task with greater answer variability and longer answers the results are broadly similar (see Table 7) except for the exact matching accuracy criterion which is too demanding because of the much larger variety of possible answers for this task.

Table 7: CoQA: the exact choice of the accuracy metric for the free-form open-book QA task has little effect on the assessment of the quality of the uncertainty measure except for the use of exact matching. For CoQA, getting an exact match is significantly harder.

| Metric | AUROC | | Accuracy |
|---|---|---|---|
| | Semantic entropy | Normalised entropy | |
| Rouge-L$(y, y') > 0.3$ | 0.7672 | 0.7533 | 0.8239 |
| Rouge-L$(y, y') > 0.5$ | 0.7379 | 0.7290 | 0.7657 |
| Rouge-1$(y, y') > 0.3$ | 0.7672 | 0.7533 | 0.8239 |
| Rouge-1$(y, y') > 0.5$ | 0.7397 | 0.7309 | 0.7677 |
| Exact matching | 0.6749 | 0.6727 | 0.6459 |

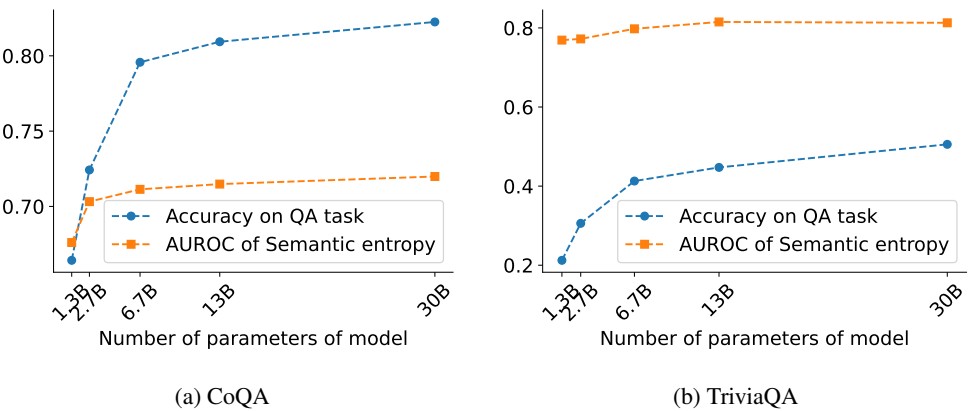

(a) CoQA  (b) TriviaQA

Figure 5: Accuracy improves with model size, as does semantic entropy's uncertainty performance. At the smallest model size, both accuracy and uncertainty diminish.

## B.4 ACCURACY ABLATIONS WITH MODEL SIZE

We confirm that increasing the model size improves the accuracy of the generations on both QA datasets (see Fig. 5a and Fig. 5b). Semantic entropy's uncertainty performance is also shown for context.

## B.5 EXAMPLE P(TRUE) FORMAT

The format of the prompt, reproduced here for convenient reference from the original source Kadavath et al. (2022), is:

```
Question:  Who was the third president of the United States?
Here are some brainstormed ideas:  James Monroe
Thomas Jefferson
John Adams
Thomas Jefferson
George Washington
Possible Answer:  James Monroe
Is the possible answer:
(A) True
(B) False
The possible answer is:
```

where the "brainstormed answers" are from the set of sampled answers $\mathbb{A}$ and P(True), i.e. the likelihood of the next token being `True` is taken as the uncertainty measure. The authors note that doing the above needs to be done in a few-shot manner and does not work well as in a zero-shot format. In our experiments, we use a few-shot prompt with 10 examples.

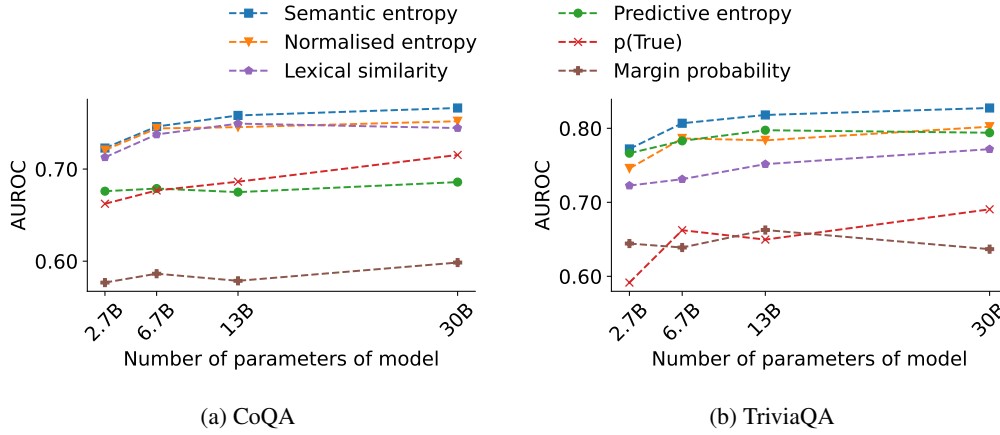

(a) CoQA          (b) TriviaQA

Figure 6: The margin probability, i.e. the difference between the likelihood of the most likely answer and the likelihood of the second most likely answer, is not very predictive of models' accuracy on CoQA open-book question answering (a) nor on TriviaQA (b). Identical to Fig. 2 with the addition of *Margin probability* which was previously omitted to avoid stretching the scale.

### B.6  MARGIN-PROBABILITY BASELINE

We additionally compare our method to the margin probability method used for neural-symbolic parsing in Lin et al. (2022b):

$$\mathcal{H}_{\mathrm{margin}}(p(\boldsymbol{y} \mid \boldsymbol{x}, \mathcal{D})) = p\left(\boldsymbol{y}^{(1)} \mid \boldsymbol{x}, \mathcal{D}\right) - p\left(\boldsymbol{y}^{(2)} \mid \boldsymbol{x}, \mathcal{D}\right),$$

where $\mathbf{y}^{(1)}$ is the top-1 beam search result and $\mathbf{y}^{(2)}$ is the top-2 beam search result.

Initially, running the method as proposed in Lin et al. (2022b) using a 13B parameter model on CoQA, we find that $\mathcal{H}_{\mathrm{margin}}$ is not very predictive of the model's accuracy on answering questions in CoQA achieving an AUROC of 0.54.

We hypothesise that two factors contribute to this poor performance. First, since this measure only looks at the difference of likelihoods, the information about the *magnitude* of the likelihood of a given answer is lost. Second—analogously to the predictive entropy—it would be important to take *semantic uncertainty* into account when computing $\mathcal{H}_{\mathrm{margin}}$. Manually inspecting model answers on CoQA, and the corresponding $\mathcal{H}_{\mathrm{margin}}$, we see that the margin between two semantically equivalent answers and two semantically distinct answers is often similar. That is, this measure does not distinguish between uncertainty between paraphrases of the same meaning (in which case the model might actually be confident about meaning of the answer), and the model's uncertainty about which semantically distinct meaning is correct.

We find that if instead of obtaining $\mathbf{y}^{(1)}$ and $\mathbf{y}^{(2)}$ by multinomial sampling (as in our other experiments) instead of by beam search, this second problem becomes less pronounced and $\mathcal{H}_{\mathrm{margin}}$ performs better while still being clearly outperformed by the other methods we study. We report our full results in Fig. 6.

Table 8: **Example of challenges for** $\mathcal{H}_{\mathrm{margin}}$. $\mathcal{H}_{\mathrm{margin}}$ does not distinguish between lexical and semantic uncertainty and thus can not distinguish cases where the model is certain about the correct answer (but uncertain about the precise formulation) as in row 1, and cases where the model is uncertain about the correct answer as in row 2. The semantic entropy correctly indicates low uncertainty in the first case and high uncertainty in the second case.

| $\mathbf{y}^{(1)}$ | $\mathbf{y}^{(2)}$ | $\mathcal{H}_{\mathrm{margin}}$ | Semantic entropy |
|---|---|---|---|
| Thomas Edison. | Edison. | 0.90 | 0.10 |
| Thomas. | George. | 0.36 | 0.87 |

