# OpenReview forum: "Semantic Uncertainty: Linguistic Invariances for Uncertainty Estimation in Natural Language Generation"
_ICLR.cc/2023/Conference — ICLR 2023 notable top 25%_

### Official Review · Reviewer_tFub · 2022-10-25

**Confidence:** 3
**Correctness:** 3
**Technical Novelty And Significance:** 3
**Empirical Novelty And Significance:** 2
**Recommendation:** 6

**Clarity, Quality, Novelty And Reproducibility:**

Nit: Sub-captions in table 1 are a bit misleading — it should mean two scenarios where there are semantically equivalent predicted answers or not.


**Strength And Weaknesses:**

Strength:
* Uncertainty estimation for generation results from large language models is an important research avenue towards improving the trustworthiness of those models.
* The motivation of measuring NLG outputs in the meaning space is quite reasonable
Experimental results showed that the proposed semantic uncertainty metric outperforms other uncertain estimation methods that operate on the surface form space of sentences instead of the meaning space, although I am not sure how to interpret the gain versus baselines using the AUC metric.

Weaknesses:
* **[Significance of Results]** Judging from AUC numbers on Figure 2 it is difficult to intuitively understand how much the proposed method is improved over the baseline, especially given the fact that results from the second-best methods on Figure 2 (a) and (b) are quite close to the proposed method. The authors also stated in Table 2 that a simple approach of counting the number of semantically distinct clusters performs quite competitively already in terms of AUROC. Does this suggest that AUROC is not a good evaluation metric since it lacks enough “resolution” to distinguish between different methods? Also, why isn’t the counting method included in Figure 2?

Could the authors consider more direct approaches to evaluate uncertainty metrics? For example, showing end-to-end answer accuracy of a decision tree: answer is from OPT if the model is certain on the input $x$ ($SE(x)$ is low), otherwise, use an oracle answer.

* **[Correctness Metric]** From Section 6, the authors used $1_{rougeL(s, s’) > 0.3}$ (typo in the original equation that uses $<0.3$?) as the metric that evaluates if a prediction
$s’$ is correct under a reference $s$. Rouge-L is a very brittle metric and doesn’t perform well when measuring semantic equivalence of short phrases. Also, the threshold of $0.3$ is chosen without further justification. Does Rouge-L and this particular threshold correlates well with human judgment of answer correctness?

* **[Models for Sentence Clustering]** The authors used bidirectional entailment as the metric to determine semantic equivalence between pairs of sentences. This task, to my knowledge, is often referred to as paraphrase identification in NLP. There is no mention of this relevant topic in section 4.2, and SoTA models for paraphrase identification literature are not used for sentence clustering in this work.

* **[Related Work]** Could the authors comment on the relationship of their work with existing research on improving generations of neural sequence-to-sequence models based on models’ uncertainty estimation (e.g., Lin et al., 2022b)? This paper is not cited.  In addition, the proposed uncertainty estimation metric in Lin et al., 2022b is not used as baseline in this paper.

`Lin et al., 2022b`: Lin, Zi, Jeremiah Zhe Liu, and Jingbo Shang. "Towards Collaborative Neural-Symbolic Graph Semantic Parsing via Uncertainty." In Findings of the Association for Computational Linguistics: ACL 2022, pp. 4160-4173. 2022.



**Summary Of The Paper:**

This paper proposes semantic uncertainty, a new uncertainty estimation metric that operates on the meaning space of natural language sentences. Semantic uncertainty accounts for the invariance of the meaning of sentences against surface syntactic or linguistic styles. Formally, semantic uncertainty of an input context is defined as the conditional entropy of the distribution over clusters of meanings conditioned on the context, and can be estimated using a model’s samples. To cluster samples into meaning groups, a bidirectional entailment model is used. Experiments are conducted on two datasets, CoQA and TrivialQA, which shows that the proposed uncertainty estimator has better correlation with the model error rate compared to some existing approaches.


**Summary Of The Review:**

Interesting idea on measuring semantic uncertainty.

---

> ### Author Response · Authors · 2022-11-11
> **Author Response to Official Review by Reviewer tFub — Part I**
>
> Dear reviewer tFub,
>
> Thank you very much for taking the time to engage with our paper so thoroughly.
>
> Below, we have tried to address all of your feedback and questions. Please take a look and let us know in case you would like additional clarification on any of these points.
>
> > **[Significance of Results]** Judging from AUC numbers on Figure 2 it is difficult to intuitively understand how much the proposed method is improved over the baseline, especially given the fact that results from the second-best methods on Figure 2 (a) and (b) are quite close to the proposed method. The authors also stated in Table 2 that a simple approach of counting the number of semantically distinct clusters performs quite competitively already in terms of AUROC. Does this suggest that AUROC is not a good evaluation metric since it lacks enough “resolution” to distinguish between different methods? Also, why isn’t the counting method included in Figure 2?
>
> We agree that it can be difficult to get a good intuition for AUC number, and we try to offer some intuitive motivation in the performance evaluation section of Section 6. AUROC is, however, a well-established and reliable metric that has been widely used in statistics and machine learning, as well as in uncertainty in natural language generation in particular (e.g. in Kadavath et al., 2022).
>
> We are confident that the AUROC does capture a real performance improvement of our method over the baselines, since we do not just report a difference in one setting: we find consistent improvements of AUROC of our methods over baselines for a range of model sizes, data sets, number of sampled model answers, and different sampling temperatures.
>
> We evaluate the AUROC of the counting method primarily as an ablation to better understand where the performance improvement of our method comes from, and we show that the counting method only explains part of the good performance of the semantic entropy. We thus do not include it in Figure 2 to avoid overloading the figure with information.
>
> > Could the authors consider more direct approaches to evaluate uncertainty metrics? For example, showing end-to-end answer accuracy of a decision tree: answer is from OPT if the model is certain on the input  ( is low), otherwise, use an oracle answer.
>
> Thank you for this suggestion.  We will prioritise implementing the additional uncertainty baseline that you refer to below but if time permits, we will try to evaluate these uncertainty measures in an additional way along the lines of your decision tree suggestion.
>
> > **[Correctness Metric]** From Section 6, the authors used $_{RougeL(s, s') > 0.3}$  (typo in the original equation that uses ?) as the metric that evaluates if a prediction $s'$ is correct under a reference $s$. Rouge-L is a very brittle metric and doesn’t perform well when measuring semantic equivalence of short phrases. Also, the threshold of is chosen without further justification. Does Rouge-L and this particular threshold correlates well with human judgment of answer correctness?
>
> Thank you for raising this point, we strongly agree that the choice of automatic evaluation metric plays an important role in our experiments. We investigate the impact of the evaluation metric on the AUROC in Table 6 in the Appendix, and find that the performance improvement of our method persists across different evaluation metrics.
>
> Following your suggestion, we add an additional ablation and manually evaluating the accuracy of 200 model answers per dataset, and compare how accurately the automatic evaluation predicts the human evaluation. We find that our automatic evaluation metric is indeed highly predictive of the human evaluation, achieving an accuracy of over 89% on both data sets. We’ve added this additional ablation in Appendix B.1 in the paper.
>
> We have fixed the typo you refer to in the paper, thank you.
>
> > **[Models for Sentence Clustering]** The authors used bidirectional entailment as the metric to determine semantic equivalence between pairs of sentences. This task, to my knowledge, is often referred to as paraphrase identification in NLP. There is no mention of this relevant topic in section 4.2, and SoTA models for paraphrase identification literature are not used for sentence clustering in this work.
>
> Thank you for pointing this out, we will extend the related work section to both connect our work better to the linguistics literature on bi-directional entailment and paraphrases, as well as the NLP literature on paraphrase detection.
>
> While there are indeed other interesting approaches to paraphrase detection, we show empirically in Appendix B.2 that our approach accurately detects semantic equivalence compare to human evaluation of semantic equivalence (92.7% and 95.3% accuracy on TriviaQA and CoQA respectively). We agree that using even stronger paraphrase identification methods from the literature to compute the semantic entropy would be an interesting avenue for future research.

---

> > ### Author Response · Authors · 2022-11-11
> > **Author Response to Official Review by Reviewer tFub — Part II**
> >
> > > **[Related Work]** Could the authors comment on the relationship of their work with existing research on improving generations of neural sequence-to-sequence models based on models’ uncertainty estimation (e.g., Lin et al., 2022b)? This paper is not cited. In addition, the proposed uncertainty estimation metric in Lin et al., 2022b is not used as baseline in this paper.
> >
> > Thank you for the recommended citation. Lin et al. 2022 seems to represent a potential use case for our method insofar as it makes use of uncertainty for a further downstream task, while our method is focused on the uncertainty measure itself. We will gladly cite this paper as an example of the value of improved uncertainty measures for downstream applications.
> >
> > We are also in the process of running additional experiments to compare our proposed method to the margin probability method described in Lin et al, 2022.
> >
> > > Nit: Sub-captions in table 1 are a bit misleading — it should mean two scenarios where there are semantically equivalent predicted answers or not.
> >
> > Thank you for pointing out that the subsections in figure 1 were unclear. We have changed them to explicitly indicate that the different tables describe two different scenarios. Please let us know in case you’d prefer an alternative labelling.

---

> > > ### Author Response · Authors · 2022-11-18
> > > **Author Follow-Up on Initial Author Response to Reviewer tFub**
> > >
> > > Dear reviewer tFub,
> > >
> > > We would like to confirm that, following your suggestion, we have implemented the margin probability baseline described in  Lin et al. 2022, see Appendix B.6 for the results. We find that, in this setting, this baseline performs less well than in other settings. We suggest that this is partially due to this measure losing information about the magnitude of answer likelihoods and because this measure also does not distinguish between lexical and semantic uncertainty. Incorporating semantic information into this measure could be an interesting avenue for future research.
> > >
> > > Following your suggestion, we have also extended the related work section with a discussion of the literature on paraphrase identification.
> > >
> > > Thank you again for taking the time to review our paper and for your valuable suggestions.

---

### Official Review · Reviewer_wSB5 · 2022-10-27

**Confidence:** 4
**Correctness:** 3
**Technical Novelty And Significance:** 3
**Empirical Novelty And Significance:** 3
**Recommendation:** 8

**Clarity, Quality, Novelty And Reproducibility:**

The paper is clear and overall easy to read. It is well written, but some useful information is spread over the document and it would be better described early on in one place (e.g. the details of  the sampling approach).

To the best of my knowledge this method has not been proposed before. However, as I point out in the previous section, the concept of assessing the variability of generated text (sampled from a language model conditioned on some input) has been used before as a feature to estimate uncertainty in NMT models. While the proposal for the estimation of semantic uncertainty in this work is different, I would expect at least some mention, and potentially even some comparison. If semantic entropy is a strong uncertainty predictor, wouldn't simple calculating the (dis)similarity of generated sentences be a strong baseline?

If authors release the code and manually annotated samples as promised (along with indications of seeds, model versions and hyperparameters chosen), it should be possible to reproduce the results with some effort.

**Strength And Weaknesses:**

The motivation for this approach is clear and well established, and the authors show that the semantic entropy can help achieve better uncertainty estimates for question answering datasets. The perform ablation tests regarding the sample size and language model size and justify well their modelling choices for the problem at hand. I enjoyed reading the paper however, I am missing a few things:

1. A better discussion of related work. The authors seem to disregard a considerable amount of work on uncertainty, unless there is a reason the focus was moved away from this work I would expect some more discussion and comparisons. As it is it is a bit harder to fully appreciate the impact of this method and place it in the space of uncertainty work. See my more detailed comment about related work at the end.
2. Some more discussion and ablation tests on the impact of generated sentence length on the computational cost and performance of the method would be nice. The datasets chosen require short sentences or single word answers, but the authors mention summarisation as a potential next step but it is unclear how the proposed semantic equivalence solution as a metric of semantic relatedness can be applied to text spans that are more complex than subject-verb—object sentences. The manual evaluation of the entailment identification method showed good performance for the datasets in question but it would be nice to see how efficient it is on longer/more complex text. As it is, I am worried that the entailment method would be significantly less accurate if applied in different scenarios.

On related work and comparisons with other work:
The mention of work in the area of uncertainty for regression is a bit outdated, especially in terms of mentioning work on "regression in low-dimensional data spaces". There are more recent works that address high-dimensional regression problems (focussing on text) such as:
-Glushkova, T., Zerva, C., Rei, R., & Martins, A. F. (2021). Uncertainty-aware machine translation evaluation. arXiv preprint arXiv:2109.06352.
-Wang, Y., Beck, D., Baldwin, T., & Verspoor, K. (2022). Uncertainty estimation and reduction of pre-trained models for text regression. Transactions of the Association for Computational Linguistics, 10, 680-696.
-Malinin, A., Chervontsev, S., Provilkov, I., & Gales, M. (2020). Regression prior networks. arXiv preprint arXiv:2006.11590

It would be important to see how the proposed method compares to these approaches instead.

Additionally, work in direct epistemic uncertainty prediction and perhaps older work in evidential deep learning should also be discussed:
- Jain, Moksh, Salem Lahlou, Hadi Nekoei, Victor Butoi, Paul Bertin, Jarrid Rector-Brooks, Maksym Korablyov, and Yoshua Bengio. "Deup: Direct epistemic uncertainty prediction." arXiv preprint arXiv:2102.08501 (2021).

On another note, I am wondering if the authors considered the lexical variability estimate that has been proposed as an uncertainty “glass-box” feature for machine translation models. The implementation is different, but I think the underlying principle is similar: sample a few sentences conditioning on the same context (in MT this is the source sentence, and they use MC dropout to sample different sentences) and then estimate the semantic diversity. The way the authors of the reviewed paper estimate semantic diversity and conceptualise eating is better established for the problem at hand, but I think it is worth comparing the two approaches and I am wondering whether the simpler approach of calculating similarity between sampled sentences yields good results as well.
Reference: Fomicheva, M., Sun, S., Yankovskaya, L., Blain, F., Guzmán, F., Fishel, M., ... & Specia, L. (2020). Unsupervised Quality Estimation for Neural Machine Translation. Transactions of the Association for Computational Linguistics, 8, 539-555.



**Summary Of The Paper:**

The authors propose a method to calculate uncertainty of language models over generated text that is conditioned on some input context. They propose to estimate "semantic entropy" i.e., entropy over clusters of semantically close groups of sampled text. The motivation behind this approach is that entropy-based metrics of uncertainty for natural language generation tasks typically consider uncertainty over tokens/token sequences without taking semantic equivalence of these into account. The authors propose to use bidirectional entailment as a way to confirm whether two sampled sentences are semantically equivalent and use this information to generate classes of semantically equivalent samples. They then calculate the entropy over these samples. They apply this method on two question answering datasets and demonstrate competitive performance compared to other methods.

**Summary Of The Review:**

Well motivated and interesting work, that requires a bit more thorough analysis of related work. I still have some questions regarding applicability to other tasks where longer/more complex sentences are expected.

---

> ### Author Response · Authors · 2022-11-11
> **Author Response to Official Review by Reviewer wSB5**
>
> Dear reviewer wSB5,
>
> Thank you very much for taking the time to engage with our paper so thoroughly.
>
> Below, we have tried to address all of your feedback and questions. Please take a look and let us know in case you would like additional clarification on any of these points.
>
> > A better discussion of related work. The authors seem to disregard a considerable amount of work on uncertainty, unless there is a reason the focus was moved away from this work I would expect some more discussion and comparisons.
>
> Thank you for raising this point. The focus of our paper is to study uncertainty for natural language generation rather than text regression. We focus on generation rather than text regression because it is both becoming increasingly important and at the same time poses unique challenges which we try to describe in Section 3 of the paper (these challenges include variable length-outputs, unequal token importance, and an extremely large output space.)
>
> We agree, however, that it makes sense to discuss related work on uncertainty for text regression more thoroughly, in particular because it might be possible to apply the concept of semantic entropy to uncertainty for text regression — an interesting direction for future work.
>
> We will thus both extend the related work section and try to make it clearer in the paper that we focus on generation rather than regression.
>
> > Some more discussion and ablation tests on the impact of generated sentence length on the computational cost and performance of the method would be nice. The datasets chosen require short sentences or single word answers, but the authors mention summarisation as a potential next step but it is unclear how the proposed semantic equivalence solution as a metric of semantic relatedness can be applied to text spans that are more complex than subject-verb—object sentences.
>
> Indeed, extending our method to other tasks such as summarisation will require more sophisticated semantic equivalence methods.
>
> Most of the literature on paraphrase identification currently focuses on identifying paraphrases between individual sentences. More advanced methods would be required to perform paraphrase identification on multi-sentence paragraphs. As soon as more advanced methods for paraphrase identification are developed, they can directly be used for our semantic entropy approach. We have added a comment about this limitation to our discussion section.
>
> Nonetheless, question-answering in itself is an important use-case for language models and we see the development of reliable uncertainty measures for this task as a valuable starting point for other free-form tasks.
>
> > On another note, I am wondering if the authors considered the lexical variability estimate that has been proposed as an uncertainty “glass-box” feature for machine translation models.
>
> Thank you for suggesting the lexical similarity-based baseline method, we will add an additional ablation to evaluate it and to compare it to our method before the end of the rebuttal period.

---

> > ### Author Response · Authors · 2022-11-18
> > **Author Follow-Up on Initial Author Response to Reviewer wSB5**
> >
> > Dear reviewer wSB5,
> >
> > We would like to confirm that, following your suggestion, we have added a baseline based on the lexical similarity of the sampled answers to our paper. We find that this is indeed a strong baseline but that it is outperformed by our semantic entropy measure (see Figures 1 and 2).  For this baseline, we use the same lexical similarity metric that we also use to evaluate model accuracy on the question-answering task, Rouge-L, and we will endeavour to additionally implement this baseline using more complicated similarity metrics such as METEOR or COMET in any camera-ready version of this paper.
> >
> > Furthermore, we have extended the related work section to include additional references to uncertainty estimation for text regression. We have also tried to clarify how our work on uncertainty for natural language generation faces fundamentally different challenges than work on uncertainty for text regression. Additionally, we extended Section 2 "Background on Uncertainty Estimation" with the work on Prior Networks and direct epistemic uncertainty prediction you refer to.
> >
> > Thank you again for taking the time to review our paper and for your valuable suggestions.

---

### Official Review · Reviewer_hFPa · 2022-10-31

**Confidence:** 4
**Correctness:** 4
**Technical Novelty And Significance:** 4
**Empirical Novelty And Significance:** 3
**Recommendation:** 8

**Clarity, Quality, Novelty And Reproducibility:**

The paper is clearly written. The quality of descriptions, justifications and support for the claims is high. The work proposes a simple, yet novel approach for measuring uncertainty in generated text. The work does seem reproducible since it doesn't need any model training and runs on off-the-shelf LLMs in an unsupervised manner. However, code was not provided as supplement so I am unable to claim this.

**Strength And Weaknesses:**

Strengths:
* This paper is very well written and has very clearly defined motivation.
* The claims made in this paper are well supported with experiments and analysis.
* The design choices have been properly justified.
* The approach is "decently obvious" and simple and I'm surprised it has not been done before.

Weaknesses and Questions:
* The last line of section 2 should have a proper citation ("While promising, these approaches need task-specific labels, additional training, and "seem to be unreliable out-of-distribution"). I don't think this work performed experiments to verify / claim this.
* The theory that semantic equivalence can be operationalized using bi-directional entailment should be grounded in linguistic theory. Please add more details about relevant linguistic work that talks about this to motivate your usage of bi-directional entailment to mean semantic equivalence more.
* In section 4.2 it is mentioned that "A sequence is part of a semantic equivalence class if it shows bidirectional equivalence with any other member of that class". Q1) Why any other? Why not a more stricter definition of all or more than half matched? However, in the discussion on computation cost just below and the pseudocode, it seems that only one sequence is used to judge if the given sequence lies in the same semantic class. Q2) Wont you need to compare with all? If you are testing if any one of them matches? Can you clarify this a bit?
* 3rd last sentence in "Computation cost" section under 4.2 is incomplete. Consequently, what is "this" in the last line (last word in 4.2)?
* Why do you use multinomial sampling? Did you try any other approach to sample?
* It is claimed in 4.4 that this method "goes some way towards addressing unequal token importance". Are there any experiments to support that (I might have missed)?
* Any insight into why (Figure 4) normalized entropy performs better than the proposed semantic entropy approach at high temperatures?

Grammar and Other suggestions:
* Be consistent in your usage of UK vs US English. Some instances in the paper have "summarization" while others have "summarisation".
* Please add links in references of tables and figures. Capitalize "Table". etc.
* Table 1 caption should be at the top of the table.
* Para 3 of 4.2 - 3rd line - Use citep instead of citet.
* Figure 2 legend "P(True) -> p(True)"


**Summary Of The Paper:**

This paper proposes an approach for uncertainty estimation in free-form text generation called "semantic entropy". The method proposed in this paper is unsupervised and can directly be applied to off-the-shelf language models. Semantic entropy provides a better uncertainty estimation than standard entropy, scales better with number of samples and performs better for model self-evaluation. This paper also provides an in depth explanation of uncertainty in free-form text generation. The method proposed first clusters sequences that mean the same thing based on bi-directional NLI. Then the average likelihoods is used to estimate uncertainty over different meanings.

**Summary Of The Review:**

The paper proposes a novel method to perform uncertainty estimation in model generated free-form text by introducing a concept of semantic entropy. The paper is very well written with clear justification for design choices and adequately supported claims and has only minor questions and clarifications.

---

> ### Author Response · Authors · 2022-11-11
> **Author Response to Official Review by Reviewer hFPa — Part I**
>
> Dear reviewer hFPa,
>
> Thank you very much for taking the time to engage with our paper so thoroughly.
>
> Below, we have tried to address all of your feedback and questions. Please take a look and let us know in case you would like additional clarification on any of these points.
>
> > The last line of section 2 should have a proper citation ("While promising, these approaches need task-specific labels, additional training, and "seem to be unreliable out-of-distribution"). I don't think this work performed experiments to verify / claim this.
>
> Thank you for pointing this out, we have adjusted the last line of section 2 to make it explicit that we are referring to existing results from the related literature rather than our own experiments: “seem to be unreliable out-of-distribution (as shown in Figures 13 and 14 in Kadvath et al. 2022)”.
>
> > The theory that semantic equivalence can be operationalized using bi-directional entailment should be grounded in linguistic theory. Please add more details about relevant linguistic work that talks about this to motivate your usage of bi-directional entailment to mean semantic equivalence more.
>
> We agree that it seems valuable to connect our bi-directional entailment method more clearly to the existing linguistic literature.
> We will extend the related work section correspondingly before the end of the rebuttal period.
>
> > In section 4.2 it is mentioned that "A sequence is part of a semantic equivalence class if it shows bidirectional equivalence with any other member of that class". Q1) Why any other? Why not a more stricter definition of all or more than half matched? However, in the discussion on computation cost just below and the pseudocode, it seems that only one sequence is used to judge if the given sequence lies in the same semantic class. Q2) Wont you need to compare with all? If you are testing if any one of them matches? Can you clarify this a bit?
>
> We will clarify that in fact if a sequence is equivalent to *any* existing member of a class then it is also equivalent to *all* of them. This is due to the transitivity of equivalence relations. That is, if sequence A is semantically equivalent to sequence B, and sequence B is semantically equivalent to sequence C, then we can infer that sequence A and sequence C are semantically equivalent without having to directly compare A and C.
>
> We make use of this transitivity property to reduce the number of comparisons between sequences we need to make while determining the semantic clusters.
>
> It is true that even though our DeBERTa-based semantic equivalence detection is highly accurate (> 92%, see Appendix B.2), it is sometimes incorrect. In practice, it might thus make sense to make more comparisons between sequences, as you suggest. We will add discussion of the trade-off between number of comparisons and quality of semantic clustering to the main body of the paper, building off our discussion already in Appendix A.1. Exploring the trade-off in detail is an interesting direction for future research.
>
> > 3rd last sentence in "Computation cost" section under 4.2 is incomplete. Consequently, what is "this" in the last line (last word in 4.2)?
>
> We’ve rewritten the last  few sentences of section 4.2 in order to clarify this: “Third, because semantic equivalence is transitive we only need to compare one member of each equivalence class to remaining sequences (see Algorithm 1). Additionally, the number of semantic clusters in our tasks is empirically quite low, see Table 2”
>
> > Why do you use multinomial sampling? Did you try any other approach to sample?
>
> In general, we tried to use the simplest version of any design choice. In this case, we chose multinomial as the simplest sampling algorithm which is also the default sampling algorithm in the HuggingFace library. In Table 4 in the appendix, we provide preliminary results on sampling using multinomial beam search sampling, where we find that this sampling method produces less diverse samples than multinomial sampling which seems to be detrimental for the semantic entropy approach. We see a more thorough investigation of different sampling methods as an interesting and important direction for future research
>
> > It is claimed in 4.4 that this method "goes some way towards addressing unequal token importance". Are there any experiments to support that (I might have missed)?
>
> We support this claim in the argument that follows the statement in the text, rather than by experiment. Our method addresses cases where two sequences mean the same thing. One way in which tokens can be unimportant is that they do not change the meaning. In these cases, with enough samples, our method should be expected to help. We have clarified the text to make it clear that this is the expected behaviour but has not been verified empirically.

---

> > ### Author Response · Authors · 2022-11-11
> > **Author Response to Official Review by Reviewer hFPa — Part II**
> >
> > > Any insight into why (Figure 4) normalized entropy performs better than the proposed semantic entropy approach at high temperatures?
> >
> > Following your observation, we manually inspected the generated answers and the performance of the semantic clustering algorithm at high temperatures on CoQA.
> >
> > We found that at these temperatures, many of the generated model answers are nonsensical combinations of words from the context that is provided for the question (e.,g, ‘ She painted herself with the same color the Farmer and others had on their horses’ on a CoQA question). While the likelihood of these sequences still seems somewhat predictive of the model’s accuracy, semantic clustering becomes very difficult and an unreliable signal for uncertainty estimation.
> >
> > We provide an  evaluation of the semantic clustering algorithm at these temperatures in Appendix A.3.
> >
> > Note that at lower temperatures, where one does get plausible and well-formed model generations, semantic entropy does clearly outperform the baselines. This finding further underlines the importance of choosing appropriate sampling hyper-parameters when using entropy-based uncertainty measures in NLG.
> >
> > > A number of minor grammatical and orthographical corrections:
> > > Grammar and Other suggestions:
> > > * Be consistent in your usage of UK vs US English. Some instances in the paper have "summarization" while others have "summarisation".
> > > * Please add links an references of tables and figures.
> > > * Capitalize "Table". Etc. Table 1 caption should be at the top of the table.
> > > * Para 3 of 4.2 - 3rd line - Use citep instead of citet.
> > > * Figure 2 legend "P(True) -> p(True)"
> >
> > Thank you for these suggestions, we have:
> > * Changed all usages of “summarization” to “summarisation”.
> > * “Please add links an references of tables and figures”, could you please clarify this? References to tables and figures in the text are currently already working links to the respectives tables and figures.
> > * Moved the caption of table 1 to the top the table.
> > * Use citep instead of citet for Para 3 of 4.2 - 3rd line
> > * Changed P(True) -> p(True)" in Figure 2.

---

> > > ### Author Response · Authors · 2022-11-18
> > > **Author Follow-Up on Initial Author Response to Reviewer hFPa**
> > >
> > > Dear reviewer hFPa,
> > >
> > > We would like to confirm that, following your suggestion, we have extended the related work section to include a discussion of bi-directional entailment in the linguistics and existing NLP literature.
> > >
> > > Thank you again for taking the time to review our paper and for your valuable suggestions.

---

> > > > ### Comment · Reviewer_hFPa · 2022-11-23
> > > > **Response to the Authors from Reviewer hFPa**
> > > >
> > > > I thank the authors for taking up my suggestions constructively and working to improve their paper. Thanks for adding additional details to the related works, clarifying your claim in section 4.4 and fixing other issues. I encourage the authors to build on this work and carry on with other suggested analysis / experiments in future work. My initial assessment is unchanged.

---

### Official Review · Reviewer_E8hD · 2022-11-03

**Confidence:** 4
**Correctness:** 4
**Technical Novelty And Significance:** 2
**Empirical Novelty And Significance:** 3
**Recommendation:** 8

**Clarity, Quality, Novelty And Reproducibility:**

- The work is mathematically well grounded. From first principles approach it makes sense to consider the semantic as the good measure of uncertainity
- The paper also empirically shows the hypothesis holds water
- This paper should be reproducible as they use all publicly available off the shelf models.


**Strength And Weaknesses:**

Strengths
- The high level idea is reasonable and beats other methods on 2 benchmark datasets
- With increase in number of samples the semantic entropy method seems to provide better auroc
- Sampling of sequence is a critical component and the paper addresses those concerns in detail.

Weakness
- The performance is shown only on two datasets.
- The author claim it is not expensive in the context of 1.5B / 30B param models. But uncertainty prediction task which adjacent to the main task of NLG, can be quite expensive if we consider smaller, production ready nlg models.
- The performance of the uncertainty prediction depends on the performance on the NLI task. And given that the clusters are formed using equivalence measure using an imperfect NLI model it would lead to clusters that are much larger indicating

**Summary Of The Paper:**

The paper is trying to produce a better measure of entropy for NLG tasks where sentences with same semantics can have really high entropy in the conventional methods as they rely on superficial measures from the exact token or sequence of tokens. The same sentence can be paraphrased in many ways and hence be equivalent to the "target".

The paper proposes a model based entropy measurement where output with same semantics would have low entropy. And this technique doesn't need any retraining.

The crux of the idea is
1. generate all the sequences based on the distribution p(s/x) ; s- sequence, x- input
2. After N sequences are computed they are clustered using an off the shelf Langauge models to produce NLI classes. Using that measure we can now cluster the sequences into C classes
3. Compute entropy SE(x) across all classes, where each class's likelihood is Sum(p(s_i/x)) Such that s_i belongs to c


They use AUROC to measure the performance of the uncertainty measurement scheme.

**Summary Of The Review:**

The paper proposes a well grounded idea to compute uncertainty in the space of NLG where the modality (natural language) of the predicted sequence allows more than one way of representing the right meaning. In language there are many ways to convey the same semantics and the NLG models would tend to distribute their output across sequences where semantics match.

So in this paper the authors propose to use entropy at the level of semantic-class rather individual queries as a measure of uncertainty. The method proves itself to be better that others.

There are some concerns around the scalability / compute this would need. And concerns around how this measure depends on the performance of the NLI model itself. And the experiments were run on just 2 datasets restricted to QA.

Nonetheless the technique seems to produce good results and paves way for future work that could address these issues.

---

> ### Author Response · Authors · 2022-11-11
> **Author Response to Official Review by Reviewer E8hD**
>
> Dear reviewer E8hD,
>
> Thank you very much for taking the time to engage with our paper so thoroughly.
> Below, we have tried to address all of your feedback and questions. Please take a look and let us know in case you would like additional clarification on any of these points during the discussion period.
>
>
> > The performance is shown only on two datasets.
>
> In our experiments, we focus on free-form question-answering data sets because they have two key advantages over other free-form generation tasks. First, automatic evaluation measures on question-answering correlate relatively well with human evaluations which is less so the case for tasks like summarisation (see e.g. Goyal et al., 2022).
> Second, off-the-shelf models perform relatively well on question-answering whereas tasks like summarisation often require resource-intensive, task-specific finetuning. In order to cover a broader range of QA settings, we make sure to evaluate our method both on open-book question answering (CoQA) and closed-book question answering (TriviaQA).
>
> Nevertheless, we agree that studying our method on a wider range of tasks seems very interesting — and are planning to do so in future work — but focus on studying question answering in this paper because it allows us to to study free-form generation while avoiding the challenges of other free-form tasks.
>
>
> Goyal, Tanya, Junyi Jessy Li, and Greg Durrett. "News Summarization and Evaluation in the Era of GPT-3." arXiv preprint arXiv:2209.12356 (2022).
>
> > The author claim it is not expensive in the context of 1.5B / 30B param models. But uncertainty prediction task which adjacent to the main task of NLG, can be quite expensive if we consider smaller, production ready nlg models.
>
> Given that uncertainty in generative large language models is a very young research area, our main goal of this paper is to establish that it is important to account for semantic equivalence in NLG uncertainty, and to propose a simple method of estimating semantic uncertainty. We therefore did not explicitly focus on the efficiency of the semantic equivalence method.
>
> We agree that in future work it would be attractive to develop more efficient methods to detect semantic equivalence than our proposed approach that uses DeBERTa-large finetuned on MNLI. Our method will further benefit from the general trend of developing more efficient NLI models. For instance,  the recent DeBERTaV3-base with only 184M parameters,  achieves comparable accuracy on MNLI (90.6/90.7) as the DeBERTa-large model (1.5B parameters) that we use in our experiments which achieves an accuracy of 91.5/91.2 on MNLI.
>
> https://huggingface.co/microsoft/deberta-v3-base
>
> https://huggingface.co/microsoft/deberta-large-mnli
>
> > The performance of the uncertainty prediction depends on the performance on the NLI task. And given that the clusters are formed using equivalence measure using an imperfect NLI model it would lead to clusters that are much larger indicating
>
> We agree that the performance of our semantic entropy method depends on the performance of our semantic equivalence detection method. In Appendix B.2, we show empirically that our NLI method is highly accurate in detecting semantic equivalence: 92.7% on TriviaQA and 95.3% on CoQA given human labels for semantic equivalence as ground truth. As the underlying models improve, we expect that our method will become more effective.

---

### Author Response · Authors · 2022-11-11
**Author Response to all Reviewers**

Dear reviewers,

Thank you all for taking the time to engage so thoroughly with our paper and for the very positive feedback on our paper:

* We were pleased that the reviewers thought that the topic of our paper is important: “Uncertainty estimation for generation results from large language models is an important research avenue” (reviewer tFub).

* We’re also happy to hear that you thought our idea is well grounded:  “The motivation for this approach is clear and well established” (reviewer wSB5);  “mathematically well-grounded” (reviewer E8hD).

* Furthermore,  we were pleased to hear that you thought our empirical results were sound: “claims made in this paper are well supported with experiments and analysis.” (reviewer hFPa);  “[they] justify well their modelling choices for the problem at hand” (reviewer wSB5); “empirically shows the hypothesis holds water” (reviewer E8hD)

 * Lastly, we were glad to hear that you enjoyed reading the paper and thought it was well written: “very well written and has very clearly defined motivation.”(reviewer hFPa); “The paper is clear and overall easy to read.” (reviewer wSB5).

Thank you also for making valuable suggestions for improvements and for pointing out things that we could have expressed more clearly. Based on your feedback we will make the following main changes:
* We will implement two additional baseline uncertainty measures as suggested by reviewers wSB5 and tFub.
* We will extend our related work section to more clearly describe how our work relates to the linguistics literature, the literature on uncertainty estimation for text regression (rather than generation which we focus on in this work), and the literature on paraphrase identification as suggested by reviewers hFPa, wSB5, tFub.

---

> ### Author Response · Authors · 2022-11-18
> **Author Follow-Up on Author Response to all Reviewers**
>
> Dear reviewers,
>
> We would like to quickly follow up to confirm that we have made the changes described above and have added them to the paper:
> * We have implemented an additional baseline based on the lexical similarity of the answers in the answer set as suggested by Reviewer wSB5. We find that lexical similarity is indeed a strong baseline but is outperformed by our semantic entropy measure, see Figures 1 and 2 in the latest version of the paper.
> * We have implemented the margin probability baseline suggested by Reviewer tFub. We find that this baseline is less predictive of model uncertainty in this setting than in previously studied settings and investigate why this might be the case in Appendix B.6.
> * Following the suggestions of reviewers hFPa, wSB5, and tFub, we have extended the related work with more discussion of the literature on uncertainty in text regression, semantic equivalence in linguistics, and paraphrase identification. These changes were both made in Section 2 “Background on Uncertainty Estimation”, as well as in Section 5 “Related work”.
>
> Thank you again for the time you took to review our paper and for your valuable feedback.

---

### Decision · Program_Chairs · 2023-01-20

**Decision:**

Accept: notable-top-25%

**Justification For Why Not Higher Score:**

[Minor - authors explained this]: It could be more impressive to evaluate the work on a wider range of tasks (two QA tasks in the current work).

**Justification For Why Not Lower Score:**

Uncertainty estimation for generation results is an important area for improving trustworthiness of large language models. Three reviewers think it is a good paper (clearly written, well motivated, interesting, well supported claims).


**Metareview: Summary, Strengths And Weaknesses:**

Summary:
The paper proposes an approach called semantic entropy, which incorporates linguistic invariances for uncertainty estimation in NLG. The proposed unsupervised method demonstrates competitive performance compared to other methods on CoQA and TrivialQA tasks.

Strength:
1. Uncertainty estimation for generation results is an important area for improving trustworthiness of models.
2. The paper is clearly written, well motivated and interesting.
3. Claims made in this paper are well supported with experiments and analysis

Weakness:
1. Most of the concerns were addressed properly by the authors’ replies.
2. It could be useful to evaluate on more than two QA datasets, as well as larger models.




**Note From Pc:**

if the above contains the word "oral" or "spotlight" please see: "oral" presentation means -> notable-top-5% and "spotlight" means -> notable-top-25%. As stated in our emails, we are disassociating presentation type from AC recommendations